# Automatic renal perfusion estimation on postoperative PCASL MRI based on deep learning image analysis and segmentation

**Anne Oyarzun-Domeño**[1,2]                                          ANNE.OYARZUN@UNAVARRA.ES
[1] *Electrical Electronics and Communications Engineering, Public University of Navarre, 31006, Pamplona, Spain.*
[2] *IdiSNA, Health Research Institute of Navarra, 31008, Spain.*

**Izaskun Cia**[1]                                                    IZASKUN.CIA@UNAVARRA.ES
**Rebeca Echeverria-Chasco**[2,3]                                     RECHEVERRIAC@UNAV.ES
[3] *Department of Radiology, Clínica Universidad de Navarra, 31008, Pamplona, Spain.*

**María A. Fernández-Seara**[2,3]                                     MFSEARA@UNAV.ES
**Paloma L. Martin-Moreno**[2,4]                                      PLMARTIN@UNAV.ES
[4] *Department of Nephrology, Clínica Universidad de Navarra, 31008, Pamplona, Spain.*

**Nuria Garcia-Fernandez**[2,4]                                       NRGARCIA@UNAV.ES
**Gorka Bastarrika**[2,3]                                             BASTARRIKA@UNAV.ES
**Javier Navallas**[1,2]                                              JAVIER.NAVALLAS@UNAVARRA.ES
**Arantxa Villanueva**[1,2,5]                                         AVILLA@UNAVARRA.ES
[5] *Institute of Smart Cities (ISC), Health Research Institute of Navarra, 31006, Pamplona, Spain.*

**Editors:** Under Review for MIDL 2023

## Abstract

Non-invasive arterial spin labeling is a magnetic resonance imaging technique that can be used for kidney transplant evaluation and perfusion estimation. This work proposes an automatic workflow for renal segmentation and perfusion estimation based on a deep learning approach and image analysis, for the postoperative evaluation of the allograft. Our method outperforms state-of-the-art results in terms of multiclass segmentation on low spatial resolution and low signal-to-noise-ratio data. Similarity coefficients above 90% are achieved for kidney, cortex, and medulla segmentation results and perfusion values within the acceptable ranges are obtained.

**Keywords:** Renal perfusion, PCASL MRI, segmentation.

## 1. Introduction

Renal transplant is the treatment of choice in patients suffering from chronic kidney disease, characterized by a progressive and irreversible loss of kidney function (Jiang and Lerman, 2019). Renal blood flow (RBF) has a great value for clinicians as it enables the identification of perfusion impairment as an emerging biomarker of transplanted renal dysfunction. The overall RBF is determined by the vasoconstriction of renal arterial tree and changes in the intrarenal vascular resistance (Field et al., 2010). Pseudo-continuous arterial spin labeling (PCASL) is a non-invasive magnetic resonance imaging (MRI) technique that allows the characterization of RBF using magnetically labeled arterial blood water spins as endogenous tracer with a combination of a train of radiofrequency pulses and slice-selective gradients

(Nery et al., 2018). It is considered an appropriate imaging technique for patients with renal dysfunction for whom the administration of contrast agents could be contraindicated (Odudu et al., 2018). RBF estimation derived from PCASL entails extra segmentation work, which is tedious and prone to error. To date, the applications of machine and deep learning models in renal MRI are scarce compared to those found for computerized tomography (CT) (Zhang et al., 2020) and high spatial resolution MRI (Klepaczko et al., 2021). We propose a fully-automated pipeline that; first performs a preliminary whole kidney segmentation by Mask R-CNN; secondly classifies the pixels within the kidney region into cortex and medulla classes; and finally estimates RBF for quantitative assessment (see Figure 1).

## 2. Experimental setup and results

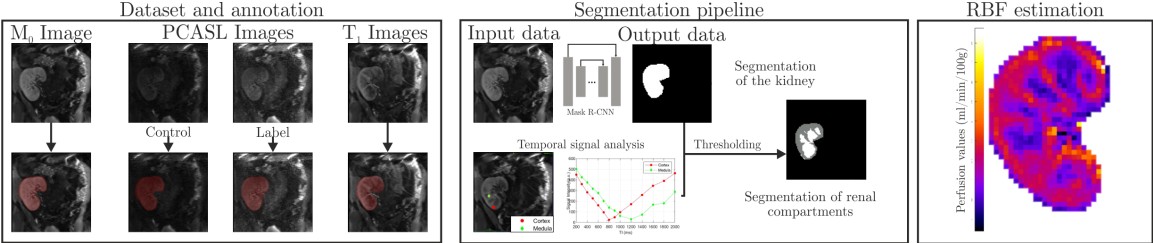

Figure 1: Automatic renal perfusion estimation pipeline.

The dataset used consists of PCASL and $T_1$-weighted (w) images from 16 transplanted patients with acquisition matrix of 96 x 96, and 3 slices. Each dataset contains a $M_0$ reference image, 25 control and labels PCASL pairs, and 14 $T_1$-w images. Binary masks encompassing the kidney, the cortex, and the medulla are used in the training and testing steps. Data augmentation and intensity normalization is used in the training process.

SEGMENTATION OF THE KIDNEY AND RENAL COMPARTMENTS

We implement the Mask R-CNN for the segmentation of the kidney on PCASL images. It consists of a two stage (Feature Pyramid Network (FPN) and a ResNet101 backbone) convolutional neural network (CNN) that generates bounding boxes and segmentation masks for each instance of an object in the image (He et al., 2018). The model is trained for 150 epochs, using supervised gradient descent optimizer, learning rate of $10^{-4}$, and pre-trained weights for MS COCO (Abdulla, 2017). We use Python 3.8 and Tensorflow on GPU NVIDIA GeForce RTX 3090. Training takes $\approx$ 120 min. We compare the performance of the Mask R-CNN against the U-Net (Ronneberger et al., 2015) and Supervised Descent Method (Xiong and De la Torre, 2013). The method proposed achieves a Dice similarity coefficient (DSC) score (mean $\pm$ standard deviation, SD) of 93.90 $\pm$ 2.00%, whereas the U-Net and SDM achieve DSC values of 87.87 $\pm$ 1.30% and 84.40 $\pm$ 11.89%, respectively. The multiclass segmentation method proposed is based on temporal information thresholding of $T_1$-w images. Due to the lack of labeled cortex and medulla tissues to train the network, we use simple image processing tools for tissue differentiation. Based on ground truth (GT) cortex and medulla annotations, we construct time-intensity curves for each tissue along

the inversion times (TI) and analyse the temporal distribution of the null points for each pixel. We note that cortical tissue attains its null point before the medulla does. Pixels within kidney masks resulted from Mask R-CNN are classified as cortex if its null point is found at $5 \leq k \leq 8$ TIs and as medulla if found at $10 \leq k \leq 13$. Unclassified pixels are designated the uncertain class. In a second stage, pixels are reclassified according to GT $T_1$ values. Instance segmentation is evaluated using a set of standard metrics: DSC, precision (PC), recall (RC), and F-measure (FM) with $\beta$ value of 2. In order to counteract the class imbalance between cortex and medulla, metrics are weighted according to the number of pixels for each class. Thus, attained RC is 89.66 ± 9.99%, PC is 91.85 ± 4.89%, FM is 89.47 ± 10.61% and DSC is 89.70 ± 10.23%.

RBF ESTIMATION

Mean cortical and medullary signals are calculated over respective tissues of subtracted control and label pairs. RBF maps are computed using the single compartment model (Nery et al., 2020). Pairwise comparisons between predicted and GT perfusion values show positive association, as the GT perfusion values increases, so does the predicted values. Obtained cortical perfusion values for proposed and GT values are: 153 ± 87 mL/min/100 g and 162 ± 70 mL/min/100 g, respectively; and medullary perfusion values of 69 ± 74 mL/min/100 g, and 67 ± 62 mL/min/100 g, respectively. Moreover, the cortical and medullary perfusion value discrepancy is 6.78% and 18.31%, respectively.

## 3. Discussion

The approach proposed leads to a reliable renal perfusion estimation. Segmentation results based on Mask R-CNN presents outstanding results, obtaining averaged DSC values above 93%, outperforming the current state-of-the-art. The segmentation performance highly depends on the intensity range of the images. Even if intensity rescale is applied, the heterogeneity of image intensity should be considered when testing with new data. As expected, the results obtained for the cortex are better than the ones extracted for the medulla compartment. The segmentation performance of medullary tissue shows higher dissimilarities between manually drawn labels and automatically achieved ones. This discrepancy is mainly caused by the mislabeling of the medulla region, which tends to be less precise than the segmentation of the cortex due to the low differentiation of interfaces and partial volume effects. The method proposed also generates an uncertain class mask in areas where the differentiation between cortex and medulla pixels is not clear, that could be processed in further steps to complete cortical and medullary masks, and indeed, the estimation of perfusion values. Regarding the estimation of renal perfusion, our work demonstrates that multiclass segmentations do have an effect on cortical and medullary RBF estimation.

## Acknowledgments

Project PC181-182 RM-RENAL, supported by the Department of University, Innovation and Digital Transformation (Government of Navarra). The author would also like to acknowledge the Department of University, Innovation and Digital Transformation for the predoctoral grant number 0011-0537-2021-000050.

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
