# OpenReview forum: "Automatic renal perfusion estimation on postoperative PCASL MRI based on deep learning image analysis and segmentation "
_MIDL.io/2023/Short_Paper_Track — MIDL 2023 Short paper track Poster_

### Official Review · Reviewer_3sC7 · 2023-04-21
**Robust application paper**

**Rating:** 7
**Confidence:** 4

**Review:**

Authors propose a three-stage pipeline to quantify renal blood flow in arterial spin labeling MRI images. First, the kidney is segmented. Second, the medulla and cortex are segmented based on temporal information. Third, renal blood flow is quantified. This is an application paper that might be of interest to those working in asl, but there is no broader methodological contribution.

Strengths

-	Well-written short paper.
-	Clear application paper with potential value for researchers in asl.
-	Results appear to be good.

Weaknesses

-	Small data set, 16 patients.
-	The segmentation step is a bit convolved. It is unclear why the authors use a Mask R-CNN model for segmentation of the kidney and then a second model for segmentation of the medulla and cortex. Can medulla and cortex not be directly segmented?
-	It remains a bit unclear what happens in the RBF estimation stage.

---

### Official Review · Reviewer_U13Z · 2023-04-24
**Limited novelty (pure application of existing methods), methodological issues, lack of comparative methods.**

**Rating:** 4
**Confidence:** 5

**Review:**

38 Automatic renal perfusion estimation on postoperative pcasl mri based on deep learning image analysis and segmentation

This abstract summarizes a pipeline that segments kidneys with r-cnn's and estimates renal blood flow. The methodological novelty is limited as it is a pure application of existing methods; redundancy in pipeline components are questionable (why using a binary segmentation of kidneys is followed by a multiclass segmentation?); lack of comparative methods. In other words, the abstract could be strenghtened by highlighting how this work advances the field with respect to the state-of-the-art methods or results. For these reasons, and situating the work with respect to other submissions, recommendation is towards rejection.